# The Role of Motion Concepts in Understanding Non-Motion Concepts

**DOI:** 10.3390/bs7040084

**Published:** 2017-12-14

**Authors:** Omid Khatin-Zadeh, Hassan Banaruee, Hooshang Khoshsima, Fernando Marmolejo-Ramos

**Affiliations:** 1Department of Language, Chabahar Maritime University, Chabahar 99717-56499, Iran; O.khatinzadeh@cmu.ac.ir; 2School of Psychology, The University of Adelaide, Adelaide, SA 5005, Australia; fernando.marmolejoramos@adelaide.edu.au

**Keywords:** metaphor, motion event, concreteness, emotions, embodiment, mathematics

## Abstract

This article discusses a specific type of metaphor in which an abstract non-motion domain is described in terms of a motion event. Abstract non-motion domains are inherently different from concrete motion domains. However, motion domains are used to describe abstract non-motion domains in many metaphors. Three main reasons are suggested for the suitability of motion events in such metaphorical descriptions. Firstly, motion events usually have high degrees of concreteness. Secondly, motion events are highly imageable. Thirdly, components of any motion event can be imagined almost simultaneously within a three-dimensional space. These three characteristics make motion events suitable domains for describing abstract non-motion domains, and facilitate the process of online comprehension throughout language processing. Extending the main point into the field of mathematics, this article discusses the process of transforming abstract mathematical problems into imageable geometric representations within the three-dimensional space. This strategy is widely used by mathematicians to solve highly abstract and complex problems.

## 1. Introduction

The nature of metaphorical domains is a subject that has been widely discussed in recent decades. In every metaphor, two domains are involved, one of which is called source and the other target. The essence of every metaphor is the understanding of the target domain in terms of the source domain [1]. Each one of these domains may include many components. In fact, the target domain is structured in terms of the source domain. It has been suggested that metaphors are understood through the mapping of relations in the source domain into relations in the target domain [2]. That is, relations among components in the source domain are mapped into relations in the target domain. A major question that has attracted a lot of attention is the nature of the domains and the way that these domains are related to each other. These domains might be abstract or concrete. Before discussing this question, we must present a clear definition of abstractness and concreteness. After looking at this point, a specific type of metaphor in which a non-motion event is described in terms of a motion event will be discussed. Among these types of metaphors, the description of emotions in terms of motions is discussed in a separate section. Extending the main point into mathematical concepts, the article discusses the process of transforming highly abstract mathematical problems into imageable geometric representations. When the abstract version of a mathematical problem is transformed into a concrete imageable (geometric) representation, the two versions of the problem are isomorphic with each other. This means they have a shared answer. Therefore, it is enough for the mathematician to solve the concrete version of the problem. After summing up the main points, the article suggests there are three reasons for the suitability of motion events (as a type of imageable domain) for describing non-motion domains. 

## 2. Abstract and Concrete Concepts

Abstract concepts have been defined as concepts that cannot be pinned down to concrete identifiable referents [3] (e.g., the concept of justice does not have a clearly perceivable referent). On the other hand, concrete concepts have identifiable and bounded referents that can clearly be perceived with our senses [3]. The Context Availability Theory and the Dual Coding Theory are two major views that have been suggested to distinguish abstract from concrete concepts. According to the Context Availability Theory [4,5], concrete concepts are strongly associated with a relatively small range of contexts, while abstract concepts are slightly associated to a much wider range of contexts [4]. Hence, this theory posits a negative correlation between concreteness on the one hand and a range width of associated contexts and strength of association on the other. The Dual Coding Theory [6,7] explains the concreteness effect with the higher imageability of concrete concepts compared to abstract concepts. According to this theory, only concrete concepts have a direct connection with images. Therefore, this theory posits a negative correlation between abstractness and imageability; that is, the more abstract a concept is, the less imageable it is. However, this intuitive relationship has been questioned by scholars [8], who have suggested that these two dimensions are correlated but not equivalent. 

Since the emergence of embodied approaches to cognition, the difference between abstract and concrete concepts has become a hot topic of discussion among researchers. According to these approaches, the understanding of concepts is influenced by perception, action, and emotion systems [9,10,11,12]. This means that understanding the concept of ‘cat’ involves the simulation or re-enactment of the perceptual, motor, and emotional experiences that are associated with seeing a cat, touching it, or hearing its sound [13,14]. Similarly, the concept of ‘grasp’ acquires its meaning through our ability to imagine, perform, and perceive the action of grasping [12]. The robust version of embodied theory of cognition assumes that imagining the action of grasping and performing it involves the same neural substrate [15]. In the discussions of embodied theories of cognition, metaphorical descriptions have attracted a lot of attention. It has been suggested that even the metaphorical use of the word ‘grasp’ leads to the activation of those sensory-motor areas that are involved in doing the action of grasping [12]. In other words, they assume that the sentence ‘*He grasped the idea*’ might activate the same sensory-motor areas that are involved in actually doing the grasping. Although embodied theories have recently been questioned by several researchers [16,17], many studies have demonstrated that the understanding of concrete concepts involves the activation of perceptual properties and emotions [3].

According to another view mainly based on the findings of neuropsychology, concrete concepts are primarily characterised by categorical similarity relations, but abstract concepts are mainly reliant on semantic associations [18,19,20,21,22]. However, this hypothesis has been challenged by the findings of several behavioural studies [23]. The ways that concrete motion events can be used to describe abstract non-motion events will be discussed throughout the following sections, which will attempt to explain how the feature of imageability makes motion events effective tools for describing abstract non-motion events such as emotions. Finally, some mathematical examples will be discussed to show how abstract mathematical problems can be understood and solved through their imageable representations.

## 3. Describing Non-Motion Events in Terms of Motion Events

It is argued here that motion events are widely used to describe non-motion events. Specifically, it is argued that a good portion of abstract concepts are described in terms of motion events. For example, when talking about *life* as a *journey*, people describe the abstract concept of life in terms of the concrete concept of journey. This concrete concept is inherently a motion event. The starting point of this journey is birth and its end point is death. The travellers on this journey are people living in this world. Ups and downs of this journey are happy and sad moments in life. Hurdles in this journey are difficulties, which people are faced with throughout life. In this metaphorical description, every abstract component of the abstract domain (life) is set in a one-to-one correspondence relationship with a concrete component in the target domain (journey).

However, the challenge is to draw a clear-cut distinction between abstract and concrete concepts. In other words, in some cases, it cannot be claimed that a given concept is one hundred percent abstract or one hundred percent concrete [3]. There are some concepts that seem to be abstract in many dimensions; however, when they are closely examined, some concrete dimensions are revealed in their semantic features. In fact, some degree of concrete association can be involved in the meaning of many concepts that seem to be abstract. It has been proposed that the relationship between concreteness and categorical and semantic associations is not binary but graded on a continuum [24]. Drawing on this argument, one view suggests that no clear dichotomy exists between concrete and abstract concepts, as concepts that are considered concrete may have an abstract component and vice versa [3]. For example, ‘Euro’ is regarded as a concrete concept. It has certain concrete characteristics such as size, colour, and weight. However, it has a value that cannot be defined by concrete perceptual characteristics [25]. In other words, although this concept is mainly concrete, there are some abstract dimensions attached to it. Based on the findings of an event-related potential study [25], it has been emphasised that although the concept of ‘banana’ is primarily concrete (having shape and colour), it has an abstract component as well, as eating it will reduce hunger [26]. One assumption is that metaphors are mainly used to describe less familiar domains in terms of familiar domains [27]. In fact, Conceptual Metaphor Theory emphasises that metaphors are usually used to describe abstract domains in terms of concrete domains [1]. Proponents of the Conceptual Metaphor Theory argue that abstract concepts are understood by mapping them into concrete domains, and this mapping guarantees their grounding [3]. For example, the abstract concepts of ‘good’ and ‘bad’ are understood in terms of right and left in the space [28] and the abstract concept of ‘power’ is understood in terms of the vertical dimension [29].

Motion events are one category of concrete domains that are widely used to describe abstract domains. There are some characteristics that make motion events suitable options for describing less familiar abstract domains. One of these characteristics is the high degree of concreteness of motion events. Every motion event usually involves a moving object, a starting point, and a path. These components can be perceived through our sensory system. In many cases of motion events, these components can be seen, heard, and touched. Another characteristic of motion events is the high degree of imageability. This feature makes motion events a suitable option for describing non-motion events. The starting and ending points of a motion event can be imagined as two points within a three-dimensional space. The path of the motion event can be imagined as a straight or non-straight line, and the manner of movement of the object can be imagined as the movement of a point along a straight or non-straight line within a three-dimensional space. Also, the main components of any motion event can be imagined almost simultaneously within the framework of a three-dimensional space. This can be particularly important for the rapid processing of information. When a non-motion event is described and understood in the form of a motion event, the components of the motion event are represented by the components of the motion event. This is a special type of simulation in which a domain is represented by another completely different domain. Although non-motion domains are very different from motion domains, simulating and representing non-motion domains using motion domains can facilitate our understanding of events. In fact, profound changes that take place through the process of simulation and representation facilitate our understanding of the less familiar target domain (the non-motion domain). The reasoning behind this capacity for understanding is that motion events are inherently easier to process and to understand. Since motion events have high degrees of concreteness and imageability, any simulation performed with reference to them can effectively facilitate the process of understanding. That is, motion domains are able to represent abstract domains in an effective manner, and make the process of understanding much easier.

## 4. Imageability of Abstract and Concrete Concepts

As previously mentioned, the Dual Coding Theory proposes that concrete concepts have a high imageability. In this theory, the assumption is that concrete concepts have direct connections with images. Among concrete concepts, those that involve movement are highly imageable, because every movement is directly perceived through the sense of vision. In fact, vision is the primary channel for perceiving the movement of elements in a concrete domain. However, abstract concepts do not have a direct visual realisation. For example, the abstract concept of religion evokes images through the mediation of concrete concepts such as church, priest, mosque, etc. [3]. With regard to imageability, another major difference between abstract and concrete concepts is that any concrete concept can be visually perceived as an entire entity. In other words, all its components can be visualised at the same time. When an entity can be visualised as an entire object, the understanding of relations among its components becomes much easier. In fact, the components, characteristics of components, and nature of relationships among components are the defining features of that entity. When all this information is visually activated during comprehension of the concept, it is said that the concept is highly imageable. The degree of imageability is dependent on two factors: the extent to which the components of the entity can be visualised, and the extent to which the object can be visualised entirely. The first factor provides minor details about the features of the object. The second factor is associated with the positions of components relative to each other. It is this factor that helps us to create a general visual skeleton of an object in our mind.

## 5. Describing Emotions in Terms of Motions

As mentioned, the degree of abstractness may vary from one concept to another. Among various concepts, emotions could be considered highly abstract, as they are related to the human psychological state. There are lots of metaphors in English and other languages that describe emotions in terms of motion events. In metaphorical description of emotions, the change of emotions is understood as a movement from one place to another. In the metaphor *He went through the roof*, the emotional changes that take place during angriness are understood as an upward movement. The starting point of this movement corresponds with the psychological state before any change, and its end point corresponds with the psychological state after emotional change. In the case of angriness, English uses an upward movement to metaphorically describe the changes that take place in emotions. In Persian, a combination of upward and downward movement is used to describe the same psychological state. Persian-speaking people say *He went up and came down* to describe the state of someone who is extremely angry. Therefore, different languages might use different directions of movement (upward, downward, horizontal, etc.) to describe the same emotional state. In the metaphor, *I am running out of patience*, a horizontal movement is used to indicate a change in emotions. In Persian, the metaphor *He went into another world* is used to talk about a profound change in feelings. In this metaphor, the direction of movement is not encoded in the motion verb. Therefore, in the metaphorical descriptions of emotions by motion events, the direction of movement might be mentioned or not. The metaphor *My heart dropped* is another example in which affective processes are understood as physical motions [30]. In this metaphor, emotions are described through a vertical downward movement.

Another point about the direction of metaphorical movement is the way that positive and negative feelings are described by motion events. In the English metaphor *He went through the roof*, a negative feeling is described by an upward movement. The same is the case with the Persian metaphor *His amperage went up*. This metaphor is used to say that someone became angry. The Persian metaphor *When I heard the news, I went into the sky* is used to describe an extreme state of happiness. These examples show that both upward and downward movements can be used to metaphorically describe negative feelings. 

## 6. Graphical Representation of Non-Imageable Concepts in Mathematics

Although some concepts are inherently non-imageable, they can still have graphical representation. The process of transforming an abstract concept (or an abstract mathematical problem) into a graphical representation of that concept (or mathematical problem) can significantly enhance our understanding of very complex abstract concepts and mathematical problems [31]. This strategy is widely used by mathematicians to solve very complex abstract problems in various branches of mathematics such as abstract algebra [32]. The key point here is the creativity of the mind in transforming an entirely abstract problem into a graphical (imageable) representation. In fact, in such a transformation, it can be said that an abstract problem has been transformed into a concrete problem. Although the abstract representation of the problem does not have a superficial similarity with the concrete representation (graphical representation), they are isomorphic at a deep conceptual level. That is, the two representations of the problem are essentially a single problem, although one is abstract and the other is concrete (graphical). In this situation, the mathematician only needs to solve the concrete version of the problem. Since the two versions of the problem are the representations of a single problem, the answer of one version is the answer of the other version. For example, there is a limitless number of vector spaces in mathematics. Vectors in three-dimensional space constitute just one of such vector spaces. Many vector spaces have a highly abstract nature. However, many of these abstract vector spaces are isomorphic with three-dimensional space. Therefore, three-dimensional space can be a good concrete mediatory tool for understanding many abstract vector spaces. In fact, three-dimensional space can represent many vector spaces, because they are isomorphic with it. In other words, many vector spaces can be understood in terms of and through the mediation of three-dimensional space. The construction of a new representation of a problem presents rich opportunities for mathematical reasoning, not only through the act of representation construction, but through a range of exploratory activities. This indicates that reasoning across representations involves the selection and coordination of entities, including aspects of constructed models [32]. 

In fact, a single abstract representation of a problem can have many concrete representations. Such cases are abundant in abstract algebra. For example, elements of a group may be real numbers, vectors, matrices, etc. [33]. However, all these groups could be isomorphic with each other at an abstract level. In other words, the concrete differences between elements of these groups do not mean that they are not isomorphic. All these groups can be isomorphic at an abstract level, although they are concretely different. This means that these groups are essentially the same at a deeper structural level, regardless of the fact that their elements are concretely different [34]. A crucial point that must not be overlooked is that sometimes mathematicians transform the geometric representation of a problem into an algebraic representation to solve it. This strategy is widely used in analytic geometry [35]. Therefore, it cannot be said that solving the geometrical (graphical) representation of a problem is always easier. It depends on the nature of the problem. While solving the geometrical representation could be easier for one problem, solving the algebraic representation could be much easier in another case. It is the creative mind that decides which representation is easier to solve, and how one representation can be transformed into another representation. The change from one representation of a problem into another representation is sometimes called modelling. Through mathematical and statistical modelling, very complex problems can be analysed in the humanities and social sciences. The use of graphical representation to analyse large amounts of data is widely used by researchers working in the humanities. 

## 7. Summary

Metaphor is a prevalent feature of human language, and motion events are one of the most common phenomena of daily activities. Hence, it would be no surprise to see the widespread use of motion events to metaphorically describe a wide range of phenomena, including non-motion events. This paper provided an explanation for why motion events can be effective tools for metaphorical descriptions of non-motion events, and even abstract concepts. Three characteristics were suggested to be behind the effectiveness of motion events in such metaphorical descriptions: high degree of concreteness, high degree of imageability, and simultaneous imageability of components within a three-dimensional space. Among these characteristics, imageability of motion events may play the most crucial role. High imageability is one of the features of concrete concepts. Therefore, it could play a major mediation role for describing unfamiliar abstract concepts. Since abstract concepts do not have a direct visual representation, the high imageability of motion events makes them a very good option for such metaphorical descriptions. This can facilitate the process of rapid and online comprehension of language. It should also be emphasised that imageability is highly reliant on the position of components within the whole system of components. In every motion event, the key elements (moving object, starting point, ending point, and path) can be visually shown in a three-dimensional space. In fact, the whole system of any motion event is highly imageable within the framework of a three-dimensional Euclidean space. In fact, geometrical representation of abstract phenomena can help us to understand those phenomena in a much more effective way. The job of a creative mind is to transform the abstract phenomena into geometrical representations, and understand those abstract phenomena through the mediation of their geometrical representations. This is a common strategy used by mathematic learners to solve complex and abstract mathematical problems [31,36,37]. Perhaps this is the reason that mathematicians are considered good abstract-thinkers. The key to this success could be their superior ability to transform abstract problems into imageable geometrical representations and solve problems through the mediation of their geometrical representations.

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
