# Peer review of "The Role of Motion Concepts in Understanding Non-Motion Concepts"

_behavsci, 2017, doi:10.3390/bs7040084_

Round 1

Reviewer 1 Report

Valuable, promising effort. Interesting conclusions. For example 

"Three characteristics were suggested to be behind the effectiveness of motion events in such metaphorical descriptions: high degree of concreteness, high degree of imageability, and simultaneous imageability of components within a three-dimensional  space."

But how can we prove that motion events are more "imageable" than the non-motion ones? Dual coding theory proposes it but that does not prove it. Also - a greater number of metaphors (both motion and non-motion) should be studied and systematically compared to draw conclusions about the motion-emotion metaphors (cf. for example Zlatev, J., Blomberg, J., & Magnusson, U. (2012). Metaphor and subjective experience: A study of motion-emotion metaphors in English, Swedish, Bulgarian, and Thai. In A. Foolen, U. M. LĂĽdtke, T. P. Racine, & J. Zlatev (Eds.), Moving ourselves, moving Others: Motion and emotion in intersubjectivity, consciousness and language (pp. 423–450). London: Equinox. (Consciousness & Emotion Book Series, 6). Retrieved from https://benjamins.com/catalog/ceb.6.17zla )  or metaphors in algebra. 

To sum up: shows a lot of promise, requires more research. 

Author Response

Dear Sir/Madam,

Three lines were added to section 4 of the manuscript to explain how motion events have high degrees of imageability compared to non-motion events. This is point emphasized by Dual Coding Theory where abstract concepts are compared with concrete concepts.

One example was added to section 5 of the manuscript to explain how motion events are employed to describe emotions. This example (taken from Zlatev, Blomberg, & Mangusson, 2012) was added to make a comparison with other examples and to emphasize the variety in the direction of movements in such descriptions.

Reviewer 2 Report

In this theoretical paper, the authors set out to show that motion concepts can help understanding non-motion concepts through metaphorical mapping. It is argued that the motion domain constitutes a suitable source domain to understand more abstract target domains, and the authors explain why this is so. They illustrate their point with mathematical thinking.

This is a theoretical paper, which means it does not present any new empirical data. For this reason my evaluation of the theoretical soundness and the clarity of the argument will be stricter than it would have been for an empirical paper.

The paper tackles an interesting problem and could potentially be insightful to researchers interested in the role of metaphors in thinking. However, in its current state it is not yet suitable for publication. For the following reasons:

1) Lack of a clear argument

In a theoretical paper it is of utmost importance that the argument is clear from the outset and remains clear throughout the paper. I found myself struggling to understand why the authors were bringing up certain points when they did, or how a paragraph related to the previous one. To illustrate, take section 2 (Abstract and concrete concepts). This section merely lists different perspectives to the abstract/concrete distinction given in the literature, without linking them or making clear which one will be crucial for the remainder of the paper. For example, the last paragraph Ls82-87 brings up yet another perspective based on neuropsychological findings and says that this view has been challenged. What should the reader do with this information? The rest of the paper is clearly not about patients, so what is the relevance? I would rather have a concluding paragraph to this section that synthesizes the main idea and naturally leads to the next paragraph, but this is not done. This is just one example in which it is not clear why something is said at a certain point, or how it relates to the previous argument, but this is a general problem of the manuscript in its current state. This leads to a lack of continuity in most of the sections of the paper.

2) Lack of supporting evidence

One function of a theoretical paper is to provide a literature review that could be useful for scholars interested in the area. On this point the paper is not satisfying. This is especially true for section 6 on mathematical thinking. Here I would have liked to see evidence that backs up the many (strong) claims of the authors. For instance, in the conclusions (section 7), the authors state that "[X] is a common strategy used by mathematicians to solve highly complex and abstract mathematical problems" (Ls245-246). But how do we know that? The authors do not cite any source that supports this claim. Is it simply based on common sense or anecdotal experience? The same objection applies to Ls 190-192: "The process of transforming an abstract concept (or an abstract mathematical problem) into a graphical representation of that concept (or mathematical problem) can significantly enhance our understanding of very complex abstract concepts and mathematical problems." I would like to see at least 2-3 references that back this up, as this is a very important claim for the paper's general argument. I am sure there *is* evidence; if so, why not cite the central scientific publications on the matter?

3) Lack of examples to illustrate the argument

While the two previous points relate to the clarity and soundness of the argument (1) and to scholarliness in citing supporting evidence (2), the last point I wish to bring up has to do with the lack of examples that illustrate the claims of the authors. For example, on Ls 197-202 talks about mathematical problems at an abstract level (no pun intended) and how they get clearer once you map them on a concrete representation. But no example is given, and so the idea is very difficult to simulate in the reader's head. To say that the two representations "are isomorphic at a deep conceptual level" (L198-199), without giving an example or specifying what is being talked about, is simply not verifiable. The authors should be more generous in providing examples that help the reader follow the argument (which, incidentally, is precisely what they claim we do when trying to understand complex relations, cf. Gentner)

In conclusion, this paper has the potential to become insightful, but the three points raised above are serious concerns that need to be addressed and resolved. Note that each concern tends to affect the paper as a whole, even if I have only given 1 or 2 examples to illustrate each point. Therefore, the paper would need thorough reworking. This could be done in many ways. One of them is for the authors to be more focused in their argument. Instead of showing that the motion domain *in general* helps thinking about the non-motion domain (a very general claim that people like Lakoff have made in several books over the years for the case of space), they could perhaps restrict their discussion to a specific range of problems, e.g. one type of problem that is especially relevant in mathematical thinking and that is well documented in the literature. My feeling is that the paper would benefit from being more focused.

Lastly, I have restricted my comments to these more general points for now, and disregarded more detailed comments. It is possible, however, that once these general points have been addressed there may be more specific points that need to be improved.

Author Response

Dear Sir/Madam

The last paragraph of section 2 was fully revised. This was done to make the text more coherent and to prepare the ground for presenting the proposals in the following sections. (first comment of reviewer 2).

One source ([31]) was added to section 6 and three sources ([31, 36, 37]) were added to section 7. This was done to back up the main proposals of the paper (second comment of reviewer 2).

One example was added to section 6 (lines 210-216). This was done to clarify the main proposal (third comment of reviewer 2).

Round 2

Reviewer 2 Report

The authors have edited the manuscript very quickly. They have partly addressed my concerns, although not in a substantial manner. In the mathematical example there is a factual error. The authors write:

"Many of vector spaces have a highly 211 abstract nature. However, all of these abstract vector spaces are isomorphic with three-dimensional 212 space."

This is simply not true. For example the 4-dimensional space is NOT isomorphic with the 3-dimensional space. One can develop intuitions about higher-dimensional spaces using 3D but saying that all vector spaces are isomorphic is simply false, and the authors should check the accuracy of what they write if their paper is about mathematical thinking, see e.g. https://mathcs.clarku.edu/~ma130/isomorphism.pdf

The authors should change the text so that their statements are not mathematically false.

I recommend accept after minor revision, but only because unfortunately after this revision I do not have very high hopes that the article will be substantially improved by further revision rounds.        

Author Response

Dear Sir/Madam,

Lines 210-216 were revised according to the comments.
